# Upgrade of ATLAS Hadronic Tile Calorimeter for the High-Luminosity LHC

Pavel Starovoitov ⬤ on behalf of the Tile Calorimeter System

Kirchhoff Institute for Physics, Im Neuenheimer Feid 227, 69120 Heidelberg, Germany; pavel.starovoitov@kip.uni-heidelberg.de

**Abstract:** The Tile Calorimeter (TileCal) is a sampling hadronic calorimeter covering the central region of the ATLAS experiment, with steel as the absorber and plastic scintillators as the active medium. The High-Luminosity phase of the LHC, delivering five times the LHC's nominal instantaneous luminosity, is expected to begin in 2029. TileCal will require new electronics to meet the requirements of a 1 MHz trigger, higher ambient radiation, and to ensure better performance under high pile-up conditions. Both the on- and off-detector TileCal electronics will be replaced during the shut-down of 2026–2028. The photomultiplier tube (PMT) signals from every TileCal cell will be digitized and sent directly to the back-end electronics, where the signals are reconstructed, stored, and sent to the first level of the trigger at a rate of 40 MHz. This will provide better precision in the calorimeter signals used by the trigger system and will allow the development of more complex trigger algorithms. The modular front-end electronics feature radiation-tolerant, commercial, off-the-shelf components and a redundant design to maintain system performance in case of single points of failure. The timing, control, and communication interface with the off-detector electronics is implemented with modern Field-Programmable Gate Arrays (FPGAs) and high-speed fiber optic links running up to 9.6 Gb/s. The TileCal upgrade program has included extensive R&D and test beam studies. A Demonstrator module with reverse compatibility with respect to the existing system was inserted in ATLAS in August 2019 for testing in actual detector conditions. The ongoing developments for on- and off-detector systems, together with expected performance characteristics and results of test-beam campaigns with the electronics prototypes, will be discussed.

**Keywords:** calorimetry; hadron calorimetry; ATLAS; tile calorimeter; large hadron collider; upgrade

## 1. Introduction

The Tile Calorimeter (TileCal) [1,2] is the central hadronic calorimeter of the ATLAS experiment [3] in the Large Hadron Collider (LHC) [4] at CERN. TileCal contributes to the measurement and reconstruction of hadrons, jets, $\tau$-leptons hadronic decays, and missing transverse momentum. It also assists in muon identification and provides inputs to the Level 1 calorimeter trigger system. TileCal is a sampling calorimeter consisting of staggered steel plates and plastic scintillating tiles that are oriented perpendicular to the beam. TileCal is divided into a long barrel, consisting of two central barrels and two extended barrels (cf Figure 1a). It covers the pseudo-rapidity range $-1.7 < \eta < 1.7$. Each barrel is segmented into 64 wedges (modules) in $\varphi$ corresponding to 0.1 granularity in $\Delta\varphi$. Each module is further segmented in the radial direction into three layers. The granularity in $\Delta\eta$ in the two innermost layers is 0.1, and it is 0.2 in the outermost layer. The segmentation in the $\eta$, $\phi$, and radial directions define the cell structure of the TileCal. In total, there are 5182 cells in 256 TileCal modules. A schematic with the TileCal module is depicted in Figure 1b. Charged particles produce light in scintillators, which is collected by wavelength shifting (WLS) fibers from two sides of each plastic tile and that then transport it to the photomultiplier tubes (PMTs). Each TileCal cell is read-out by two PMTs to provide signal redundancy and to improve the energy resolution.

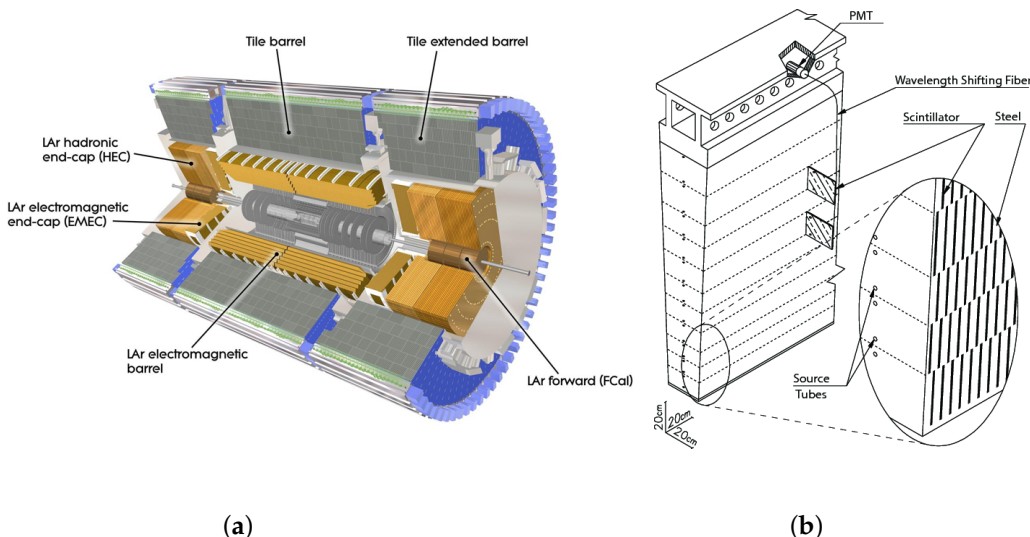

(**a**)                                                                    (**b**)

**Figure 1.** (**a**) Cut-away view of the ATLAS calorimeter system. The Tile Calorimeter, consisting of one barrel and two extended barrels, is shown with gray color in the outermost part of the picture. (**b**) The mechanical structure and optical readout of a single Tile Calorimeter module. The scintillating tiles are oriented normally to the beam-line and read-out by fibers in the radial direction.

In the present system, the PMT signals are digitized at a frequency of 40 MHz. The digital samples are stored in on-detector pipeline memories. At the same time, the trigger analog boards sum the PMT pulses into pseudo-projective $0.1 \times 0.1$ in $\eta-\varphi$ space trigger towers and send them to the Level-1 calorimeter trigger system. Once the event is accepted by the hardware-based Level-1 trigger system, it is extracted from pipelines and sent for further processing. The Level-1 trigger system processes events at the maximal rate of 100 kHz. The software-based High-Level Trigger system (HLT) refines the events selected at the Level 1 trigger and transfers them to local storage at about a 1 kHz rate.

The datasets collected by the LHC experiments during the Run-1 and Run-2 operations allowed for testing the Standard Model (SM) of particle physics with remarkable precision up to unprecedented energy scales in the sectors of strong and electroweak interactions, heavy flavor physics, and Higgs boson production. The High-Luminosity upgrade of the LHC (HL-LHC) [5] will provide an instantaneous luminosity that is five times larger than the nominal LHC one, with a goal to collect 4000 fb$^{-1}$ integrated luminosity by the end of the HL-LHC data taking. This dataset manifests a new precision frontier in the Higgs sector, opens up unique opportunities to study rare production processes and offers a chance to probe for Beyond the Standard Model physics with unprecedented sensitivity. Among most important are the measurements of the Higgs boson couplings to a percent accuracy and its self-coupling, studies of the di-Higgs production, and decays in Higgs boson couplings to invisible states, as well as the production of three massive vector bosons, searches for lepton flavor violation, and measurements of four top-quarks production mechanisms.

Such an ambitious physics program implies a high event rate and requires excellent read-out capabilities of the sub-detectors, supreme selectivity and functionality of the trigger system, and ultimate performance of the data-acquisition system (DAQ). The high-luminosity conditions impose significant challenges for the detector, trigger, and data-acquisition systems. Approximately 200 simultaneous proton–proton collisions will be produced in every bunch crossing on average, leading to a significant increase in the particle flux in the detector. TileCal on-detector electronics will receive up to about 160 Gy of total ionizing dose (TID) during the full HL-LHC data taking, which is an order of magnitude larger than one in Run-2. The HL-LHC Tile Calorimeter must satisfy the broad physics program goals and withstand extremely challenging conditions during a decade-long data taking.

In the HL-LHC system, the digitized PMT signals are sent to the off-detector electronics, where the energy reconstruction is performed for every TileCal cell. Trigger objects of different granularity, i.e., $0.1 \times 0.1$ and $0.2 \times 0.2$ in $\eta-\varphi$ space trigger towers and individual cells with energy depositions above a certain threshold (e.g., cells compatible with minimum ionizing particle energy losses for muon triggering), are built and sent to the L0/L1 Calo, L0Muon, and Global trigger systems for further processing [6]. Therefore, the full readout system has to be replaced in order to handle the increase in the data rate, to enhance radiation tolerance of the on-detector electronics, and to be compatible with the fully digital ATLAS trigger and data-acquisition system for the HL-LHC.

In addition, 10% of the most exposed PMTs will be replaced by new PMTs, while the remaining optics will be retained. The higher radiation levels also require the redesign of the low-voltage (LV) and high-voltage (HV) power distribution and regulation systems.

The HL-LHC will start the proton–proton collisions in 2029. The ATLAS experiment has started the design and construction of the upgraded detector [7,8] to fully exploit the physics potential of the HL-LHC dataset.

This contribution presents the Tile Calorimeter upgrade program [9] and discusses its current status. It is organized as follows. In Section 2, the new design of the TileCal mechanical structure is described, and the problem of optical components aging is addressed in Section 3. The upgrades of both the on-detector and off-detector electronics are discussed in Section 4. Sections 5 and 6 describe the low-voltage and high-voltage power supply systems, respectively. Finally, the measurements of the hadron response performed in test-beam studies using the upgraded electronics are presented in Section 7. The results are summarized in Section 8.

## 2. Mechanical Structure of TileCal

TileCal is organized into three cylindrical volumes (barrels), one Long Barrel (LB), consisting of two central barrels, and two Extended Barrels (EB), centered along the beam axis. Each barrel consists of 64 identical modules.

An EB module is read-out by 32 PMTs housed in a super-drawer (SD) together with the associated electronics. Super-drawers are located at the outer radius of the calorimeter. An LB module is read out by $2 \times 45$ PMTs that are placed into two super-drawers together with related on-detector electronics. The total number of PMTs is 9852, and almost all calorimeter cells are read-out by two PMTs for redundancy, while only a few cells are read-out using the single-PMT scheme.

The SDs are constructed of mechanically linked Mini-Drawers (MD). They hold the PMTs and on-detector electronics. Each MD can hold up to 12 PMTs and is electrically independent from the adjacent MDs. Four MDs are linked to create one LB super-drawer, while only three MDs are required to build one EB super-drawer. In EB super-drawers, two additional micro-drawers hold PMTs in the right position. The micro-drawers do not hold digital electronics. This modular design was chosen to facilitate handling during the installation and maintenance. The MDs are produced from aluminum and high-density, fire-retardant polyethylene. Cooling of the electronics is ensured via a cooling bridge and a water channel inside the aluminum frame. Following extensive prototyping and tests in beam at CERN, the design has been finalized, and the production will be completed by the end of 2022.

## 3. Aging of the Optics and Long-Term Robustness Tests of New PMTs

Most of the optics elements in the TileCal cells cannot be replaced without dismounting the entire ATLAS detector and a complete disassembling of the TileCal. Dedicated analysis of the TileCal performance and aging during the Run-1 and Run-2 data-taking periods [9] indicate that the aging of these elements will not significantly impact the TileCal's performance throughout the full HL-LHC data taking. Therefore, neither scintillating tiles nor the WLS fibers will be replaced towards the HL-LHC upgrade.

The PMT response varies over time during data taking because it is affected by variations in the photo-cathode quantum efficiency and dynode multiplication gain due to the integrated anode charge over the running period. To ensure excellent detector performance during the physics collisions, the PMT response is monitored every 2–3 days with the Laser calibration system, sending laser light pulses to all the PMTs. Using the Laser calibration system, the PMT response variation during Run-1 and Run-2 was studied in great detail [10,11]. The PMT response is characterized by down-drifts and up-drifts (response recovery). The down-drifts coincide with the collision periods, while the recoveries occur during the technical stops. The observed down-drifts mostly affect the PMTs that are reading out the TileCal's innermost layer [12]. The estimated maximum integrated anode change at the end of the HL-LHC data taking is around 600 C for most exposed TileCal cells.

Since the degradation in the PMT response depends on the amount of integrated anode charge, the study of the PMT response variation as a function of the integrated anode charge is performed to understand the PMT's performance in the HL-LHC's conditions. In this study, PMTs are excited using the laser light, while a light-emitting diode (LED) is used to generate a charge that is integrated by the PMTs. All PMTs are equipped with high-voltage active dividers to ensure the PMTs' linearity over a wide range of anode currents. The results of the study are shown in Figure 2. Red triangular points represent the average response of PMTs model Hamamatsu R7877 dismounted from TileCal detector in February 2017. After being dismounted, these PMTs already integrated up to 40 C in previous laboratory tests. Black circular points represent the average response of PMTs model Hamamatsu R11187, an evolution of model Hamamatsu R7877. Over time, the number of PMTs in a given data bin can vary, due to the different charge integrated by a given PMT. Missing data points correspond to a period of time in Fall 2021, when many interruptions in data taking had taken place with interrupting the charge integration.

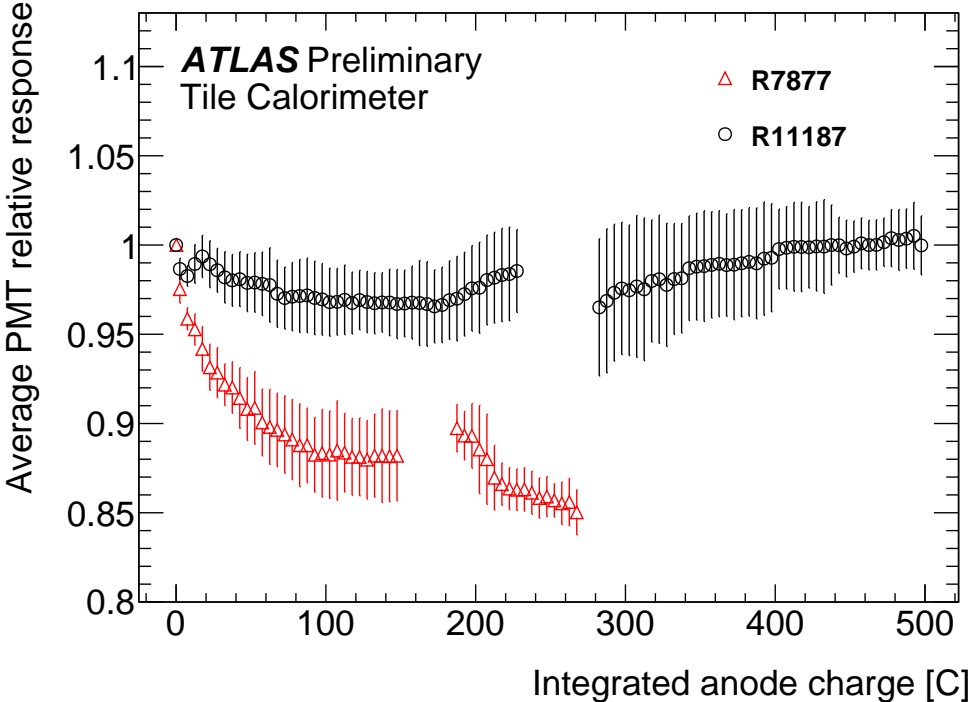

**Figure 2.** The PMT response as a function of the integrated anode charge. The response is plotted in the units of PMT response measured at zero integrated anode charge. Two different PMT models are shown. Each point is evaluated as the average of the response of the set of PMTs with the same integrated charge. The error bar corresponds to the RMS of the PMT response over the set.

With more than 400 C (several PMTs are already at 500 C) of integrated anode charge, all but a single PMT show excellent performance, retaining full response [12,13]. Both tested PMT models match the Tile Calorimeter performance requirements through the HL-LHC's data taking. Based on the detailed studies, approximately 10% of all PMTs, corresponding to the most exposed cells, will be replaced within the HL-LHC upgrade program.

## 4. Read-Out Electronics

The TileCal read-out electronics are divided into two main domains: on-detector electronics that are housed inside the SDs and must pass stringent requirements on the radiation hardness and off-detector electronics that are located in underground counting rooms about 100 m away from the ATLAS detector. There are no requirements on the radiation hardness for the off-detector electronics. The TileCal read-out scheme for the HL-LHC upgrade is presented in Figure 3.

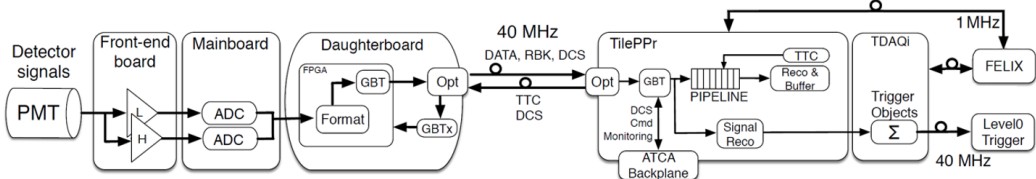

**Figure 3.** The TileCal Phase-2 upgrade read-out scheme. The on-detector electronics section (left diagram side) with the PMTs, FENICS, Main Board and Daughter Board is connected by the long fibers to the off-detector electronics section (right diagram side) with the Tile PreProcessor. The TDAQi module is optically interfaced to the trigger and DAQ systems.

### 4.1. On-Detector Electronics

The TileCal HL-LHC on-detector electronics chain is housed inside the mini-drawers. Each mini-drawer contains up to 12 PMT blocks, each with a light mixer, a PMT, its active high-voltage divider base, and an amplifier/shaper card (FENICS). The PMT block is connected to a Main Board that provides a low voltage and controls to FENICS cards, digitizes their signals and routes the data to the link Daughter Board that handles all high-speed communication with the back-end electronics. Four major constituents of the on-detector electronics are discussed in this section.

### 4.1.1. PMT Blocks

The PMTs are located inside metal cylinders that include active HV-dividers and analog read-out boards, FENICS. The FENICS boards shape and amplify the PMTs pulses and are based on the commercially available off-the-shelf (COTS) components [14]. Two types of signal processing are implemented. The "fast readout" for physics data taking operates in two different gains in order to cover the dynamic range from 200 fC up to 1000 pC to measure energies of a few hundreds MeV in a single particle or a multi-TeV hadronic jet. The "integrator read-out" integrates the PMT current for the calibration of the calorimeter with a $^{137}$Cs source. It also provides a relative measurement of the accelerator luminosity at the ATLAS interaction point. The integrator read-out uses six different gains to precisely cover eight orders of magnitude in luminosity measurements, ranging from a dedicated van der Meer scan to physics collisions data. The FENICS board is also able to inject a precise charge and to measure the conversion from pC to ADC counts. The radiation qualification of FENICS, along with validation of the design, has been completed, and pre-production has been accomplished in Summer 2022.

### 4.1.2. High-Voltage Active Dividers

In order to ensure a 1% precision on the energy scale in TileCal, the high voltage fed to the eight-dynode PMTs must be stable within 0.5 V in the range 600–900 V. During the HL-LHC data taking, the PMT current can reach 40 µA in the most exposed TileCal

cells. Therefore, active dividers are necessary to maintain a better than 1% linearity. The linearity of the active–divider–PMT system was successfully tested in a dedicated set up that emulates pulses from physics collisions on top of a continuous soft activity from pile-up. The active dividers were fully validated in radiation tests, and production has started.

### 4.1.3. Main Board

One Main Board installed in each MD digitizes the data from up to 12 PMTs. It is based on the COTS components. The fast-read out uses 24 12-bit ADCs operated at 40 MHz, while for the integrator readout, it uses 12 16-bit SAR ADCs. The Main Board routes the data to the Daughter Board and provides digital control of the FENICS to configure it for either a calibration run or for the physics data taking. For robustness and redundancy, both the Main Board and the Daughter Board are designed to have two electrically independent sides. As each calorimeter cell is read out by two PMTs, these two PMTs are connected to two different sides of the Main Board [14]. The Main Board was fully qualified for the expected radiation environment in the HL-LHC and has already entered the production stage.

### 4.1.4. Daughter Board

There is one link Daughter Board (DB) per MD. It is responsible for the high-speed communication (4.8/9.6 Gbps) with the off-detector electronics. The Daughter Board sends precision data as well as the slow control and monitoring data from the on-detector electronics. The Daughter Board receives the LHC clock and distributes it to the on-detector electronics and exchanges configuration and sends control commands to the front-end boards [15]. Each Daughter Board uses 2 Kintex Ultrascale FPGAs. The Daughter Board design has been extensively tested and is validated in several test-beam campaigns in the H8 line in the North Area of CERN. The active components of the DB, such as optical transceivers, FPGA, and isolation amplifiers, are having their radiation hardness evaluated. The design will be finalized in 2023.

### *4.2. Off-Detector Electronics*

During HL-LHC operation, the PMT digital samples for every bunch crossing will be transferred from the Daughter Board to the off-detector electronics, where the data will be reconstructed at 40 MHz frequency.

The reconstructed information includes calibrated energy and time per calorimeter cell or group of cells depending on the trigger system. The energy and time of the PMT pulses are calculated using the digital filters. The trigger primitives are sent to the Level-1 trigger system, while data are stored in pipeline buffers waiting for a trigger decision. The TileCal off-detector electronics are hosted in four ATCA crates, and eight PreProcessor (PPr) blades are installed per crate. In order to have more flexibility and reduce the complexity of the designs, the PPr functionality is split between three different boards. The basic interface with the ATCA backplane and the power distribution is located on the custom ATCA Carrier Base Board (ACBB). The interface with the front-end electronics, pipeline buffering and signal reconstruction are implemented in the Compact Processing Module (CPM). Each ACBB hosts four slots to install CPM boards [16]. The construction of the trigger primitives and the interfaces with the L0/L1 trigger system and Front End LInk eXchange [17] (FELIX) systems are performed in the Trigger and DAQ interface (TDAQi) system [18]. The diagram of a single PPr blade is presented in Figure 4.

The PPr system provides multiple copies of the data for those parts of the detector where the area used in the trigger algorithms overlap. After the trigger decision, the selected data events are transferred to the FELIX system, which is the core element of the central ATLAS DAQ system.

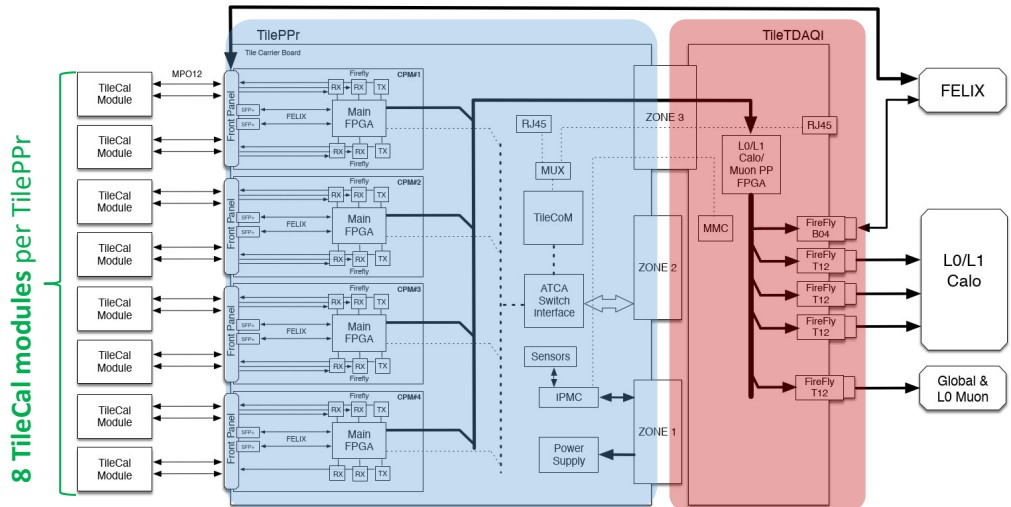

**Figure 4.** Block diagram of the TileCal Preprocessor. A complete PPr system is able to process the data from eight TileCal super-drawers.

## 5. Low1

A three-stage, low-voltage system is presented in Figure 5. Low-voltage power supplies (LVPS) are placed on-detector to convert 200 VDC to 10 VDC as required for the on-detector electronics. The LVPS design is based on the COTS components. However, due to their position in a MD, the LVPS are the TileCal components most exposed to radiation. Following a series of extensive irradiation test campaigns, radiation hard components were identified.

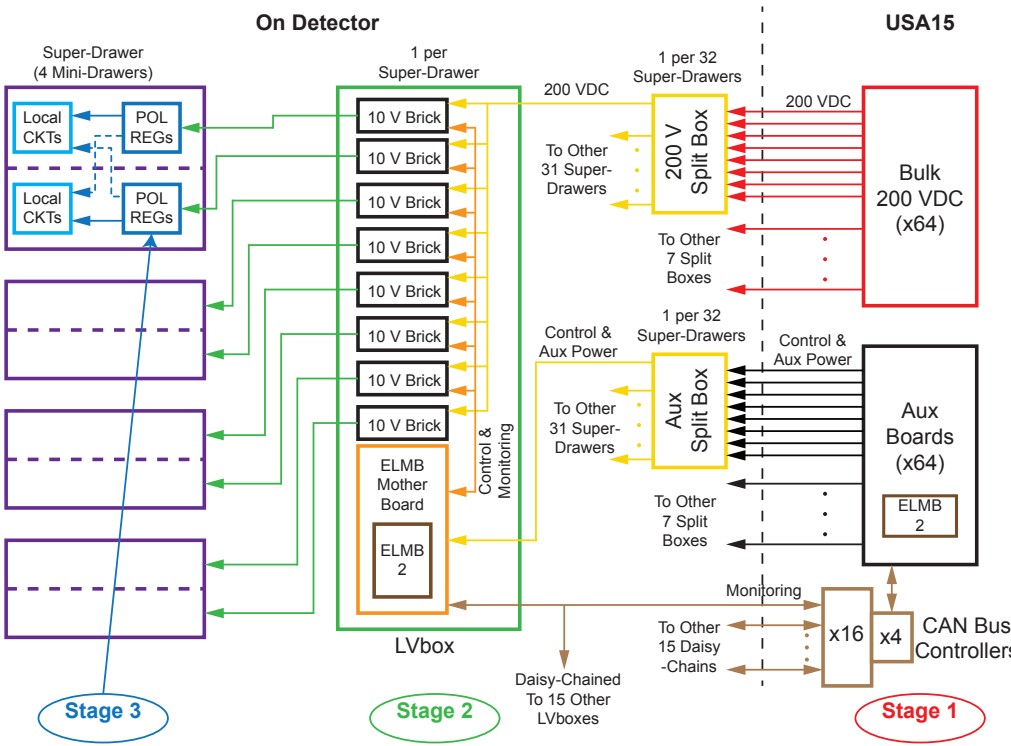

**Figure 5.** The block diagram of the TileCal low-voltage distribution system for the HL-LHC upgrade. The "10V bricks" stand for 200 VDC to 10 VDC converters. Stage 3 is implemented directly on the front-end electronics boards.

The monitoring and control functions of the LVPS were separated to improve robustness. The monitoring functions are ensured by the radiation hard version of the Embedded Local Monitor Board (ELMB). The new ELMB2 board is functionally equivalent to the legacy ELMB.

The on-detector part of the ON/OFF/ENABLE control of each low-voltage converter is built using a passive system implemented on a so-called ELMB-motherboard. It uses a tri-state voltage levels to minimize the necessary cabling. The control system allows for individual control of a single MD LVPS. In addition, each Main Board and Daughter Board half-board is powered by a separate power converter to minimize the impact of a single component failure on the system's performance. The ELMB-motherboard also routes power to the ELMB2. The production of the ELMB-motherboard will be completed in Summer 2022.

Point-of-load regulators located directly on the Main Board and the Daughter Board make up the third stage of the LV system. A full vertical slice of the LV system was built and tested at CERN in the period from 2019 to 2022. It demonstrated the required functionalities and robustness of all three stages of the LV system [19]. The LVPS boards will go into pre-production in Fall 2022.

## 6. High-Voltage Distribution and Regulation

The High-Voltage (HV) TileCal HL-LHC upgrade system consists of HV-remote boards and HV-supplies boards, located far from the detector in custom-designed HV-crates, connected to the on-detector components by ∼100 m long HV-cables [20]. Inside the detector, the passive HV-bus boards are used to bring the HV individually to each PMT located inside a mini-drawer. The HV-remote boards separately regulate each HV channel using a dedicated regulation loop. This regulation loop is a simplified version of the legacy one [21]. As in the HL-LHC, this board is located off-detector and is not affected by the radiation damage. The regulation scheme controls the high voltage in each channel to within 0.5 V. The total number of HV-remote boards is 256.

The main functional improvement in the HL-LHC HV system compared to the legacy one is an addition of ON/OFF control for each group of four channels complemented by a jumper for each individual channel.

A single input HV, either −830 V or −950 V, is provided for every group of 24 channels, while the voltage of each channel is regulated individually in a range of 360 V. The primary HV is provided by the Hamamatsu C12446-12 modules mounted on the HV-supply boards, and two primary HV inputs are used to provide HV for 48 (32) channels in a Long (Extended) Barrel module.

The HV-bus board is the only component of the HV distribution system that is installed on the detector. These boards are fully passive and have four layers to ensure the protection of the tracks with high-voltage in the inner layers of the board. The TileCal HL-LHC upgrade HV distribution system is shown in Figure 6.

The latest prototypes of the full HV system components were successfully used in the TileCal test-beam campaign of June 2022.

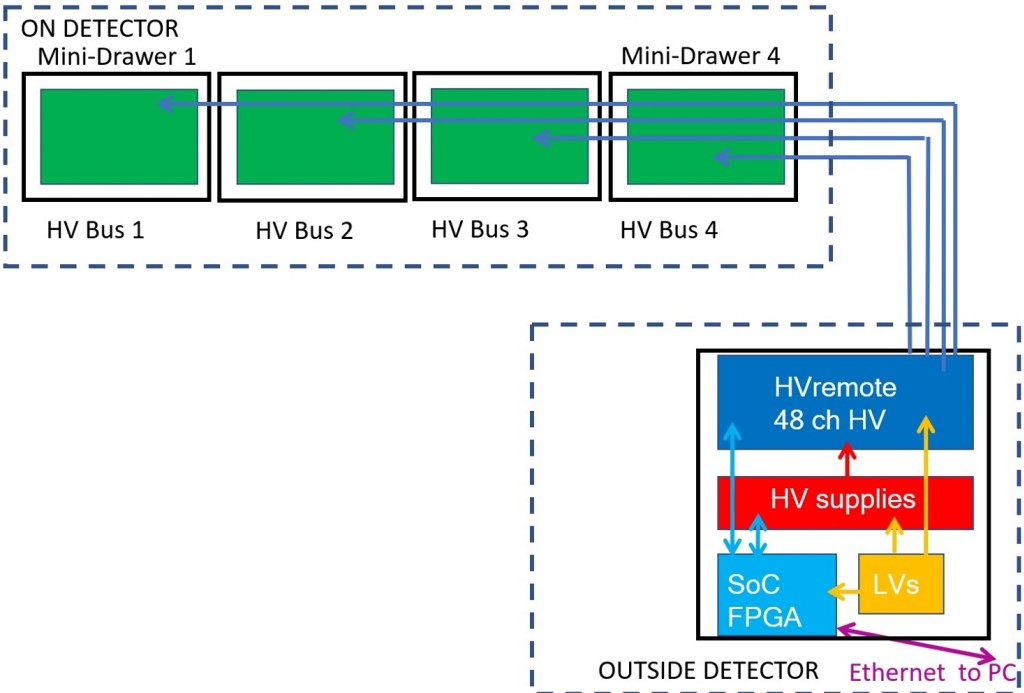

**Figure 6.** The high-voltage power supply system scheme for the Phase-2 upgrade. The regulation system is remote, located far from the detector, and a large number of 100 m long cables bring the HV to the detector modules. Inside the detector, the HV is distributed to 4 (3) mini-drawers in the Long Barrel (Extended Barrel) modules.

## 7. Test-Beam

Several test-beam campaigns were carried out at CERN with the HL-LHC TileCal upgrade electronics. Three Tile Calorimeter modules, two LBs and one EB, were exposed to hadron (pion, kaon and proton) beams with energies, $E_{\text{beam}}$, ranging from 16 to 30 GeV [22].

The MDs were equipped with FENICS cards, Main-Boards and Daughter Boards. The on-detector electronics were powered with the pre-production version of the LVPS, and the latest prototypes of the HV system were used to operate the PMTs. The on-detector electronics were configured through the upgrade version of the TileCal PreProcessor, which was also used to take both physics and calibration data. In addition, one module was equipped with a combination of the upgrade and legacy electronics. The aim was to study the performance of the different versions of upgrade electronics and to obtain a direct comparison with the legacy system. The good performance of the new electronics was demonstrated during these test beam campaigns.

In addition to the tests of the latest electronics prototypes, the test-beam campaigns are used to study the TileCal response to hadrons with different energies. The hadron identification is performed using a system of Cerenkov counters [22]. The energy deposited by an incident particle, $E_{\text{c}}^{\text{raw}}$, is reconstructed as a sum of energy depositions in all calorimeter cells with energies twice as large as the noise threshold, $\sigma_{\text{noise}}$. The noise is determined in events collected between beam bursts using a random trigger, typically $\sigma_{\text{noise}} \sim 30$ MeV. The energy response is defined as a ratio of the mean deposited energy to the beam energy, $\mathcal{R} = \langle E_{\text{c}}^{\text{raw}} \rangle / E_{\text{beam}}$. The measurements of the Tile Calorimeter response to positive pions, kaons and protons as a function of beam energy are presented in Figure 7. The measurements are also compared to the predictions obtained with the Geant4-based [23,24] ATLAS simulation toolkit [25] of the TileCal test-beam experimental setup.

The energy response results for pions, kaons and protons determined in data and in simulated events agree within the uncertainties. On average, the differences between the data and Monte Carlo simulations of the energy response was found to be 1.1% with an average total uncertainty in the energy response determination of 1.4%.

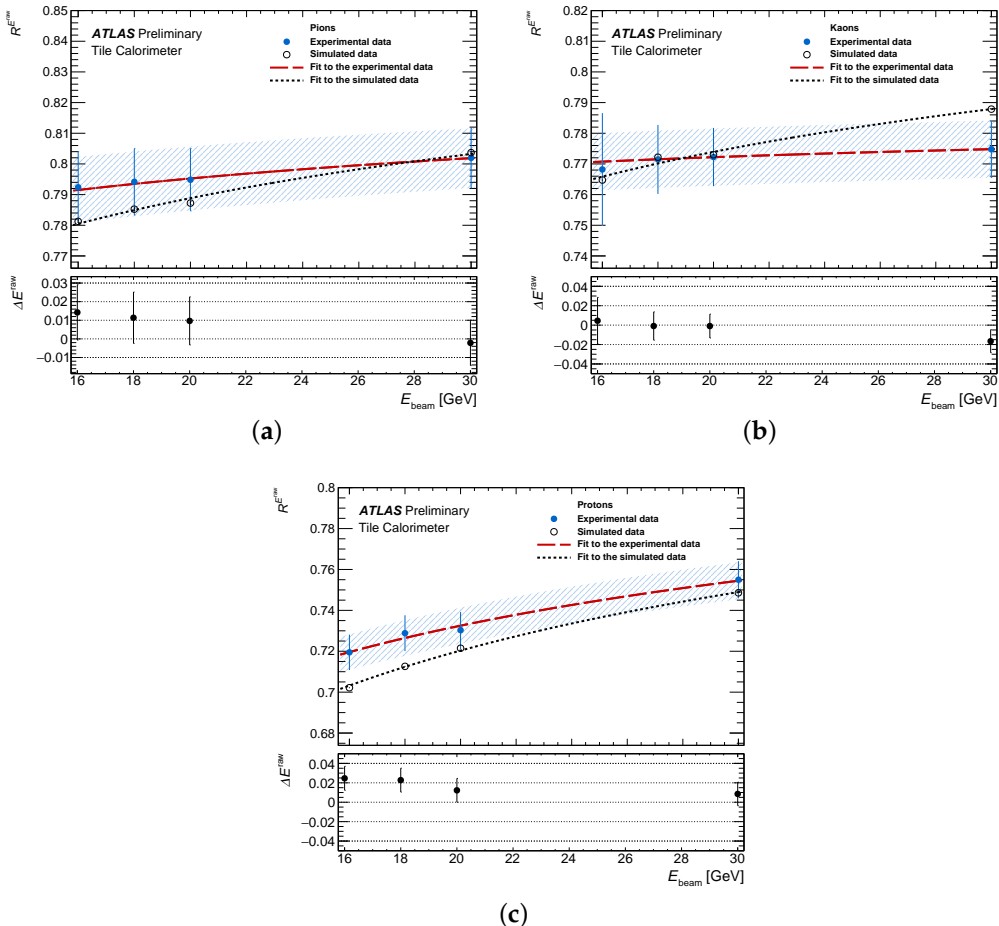

**Figure 7.** Energy response normalized to incident beam energy measured (blue dots) and predicted by MC simulation (black circles) as a function of beam energy obtained in the case of (**a**) pion, (**b**) kaon and (**c**) proton beams. The details of the fits to data (red line) and MC simulation (black line) are explained in Ref. [22]. The experimental uncertainties include statistical and systematic effects combined in quadrature. Simulated results show only statistical uncertainty.

## 8. Conclusions

The HL-LHC will open unprecedented possibilities to test the Standard Model of particle physics and take a glimpse on a possible manifestation of new phenomena or production of unknown particles. In order to withstand the tough radiation environment and exigent particle flux conditions and to provide superlative detector performance, the upgrade program was launched by the ATLAS experiment. It requires a full replacement of TileCal on-detector and off-detector electronics and the development of new approaches to power the detector electronics. The design of the ATLAS Tile Calorimeter upgrade for HL-LHC is essentially complete; all parts of the system have been prototyped and validated in standalone test-benches, as well as in integration tests together with other parts of the TileCal upgrade project. The radiation tests of the active components used for the pre-production and production series have been proven to be sufficiently radiation hard to resist estimated particle fluxes and guarantee the design performance. Several parts of the on-detector electronics are already in the final production stage, while the rest are in the pre-production phase. System-level tests for the low-voltage distribution system and the high-voltage power supply system have been successfully completed. Some elements of the LVPS system are already being produced, while the rest of LVPS and the high-voltage power supply system are entering the pre-production stage. Several test-beam campaigns were organized throughout the last decade, where the latest prototypes of various TileCal subsystems were simultaneously validated in the real data-taking environment. The

accumulated data are used to constrain the modeling of a hadron interaction in matter in detector simulation frameworks.

The TileCal upgrade project is on schedule for the system on-surface integration in 2024 and installation in ATLAS by the end of 2026.

**Funding:** This research received no external funding.

**Data Availability Statement:** Data sharing is not applicable to this article.

**Conflicts of Interest:** The author declares no conflict of interest.

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
