# Peer review of "Upgrade of ATLAS Hadronic Tile Calorimeter for the High-Luminosity LHC"

_instruments, doi:10.3390/instruments6040054_

Round 1
Reviewer 1 Report
I have only minor textual comments below.
L2 – steel as absorber-> steel as the absorber
L2 – scintillators as active medium -> scintillators a the active medium
L3 – of LHC – of the LHC
L7 – PMT is later defined in text. Since you define FPGA in abstract, try to be consistent with definintion of PMT as well – either define both in abstract or both in text.
L11 – and redundant – and a redundant
L39: at the 40 MHz frequency – at a frequency of 40 MHz
L43: It might be helpful to specify that the L1 is hardware based, and HLT software based.
L47: It would be good to include this motivation paragraph earlier in the introduction if possible
L54: Thus, around -> Approximately
L56: How does the radiation dosage compare with the LHC?
L57: full readout system -> the full readout system
L59: Please define DAQ the first time you refer to it in the text earlier on
L60: the 10% most exposed - > the 10% of the most irradiated
L61: kept ->retained
L76: While only few cells….: this second part of the sentence is redundant and can be removed
L91: Is there a reference for the dedicated analysis?
L113: around -> approximately. The most exposed ones -> corresponding to the most exposed ones
L167: it also exchanges ->and exchanges
L171: are being examined the radiation hardness -> are having their radiation hardness evaluated
L174: in the HL-LHC -> During HL-LHC operation. Are transferred from -> will be transferred from. Data are reconstructed -> data will be reconstructed.
L230: about
L234: on-detector -> on the detector.
L243: Don’t need the “There,”, can just start with Three TileCal…
Figure 7: The legend is difficult to read, is it possible to enlarge?
L259: of mean deposited energy -> of the mean deposited energy
L171: to withstand tough -> to withstand the tough
L290: this is the first time Point 1 is mentioned, it might be better to just say “and installation in ATLAS”
L290: Join the last sentence to the concluding paragraph.
Author Response
Dear Reviewer,
thanks a lot for the careful reading of my article. I have modified the text according to your suggestions.
The only exception is your comment on L47. This paragraph describes the transition between the current and upgrade systems rather than motivation. I have added few sentences to make it more clear.
Best regards
Reviewer 2 Report
Dear Author,
in my opinion your nice and clear paper is ready for publication.
I congratulate you for it.
I just have a couple of very minor comments/corrections:
- line 34: the number of ‘256 TileCal Modules’ looks not appropriate, given
in the above description (but also later in lines 69-70) you quote that
TileCal is organized into three barrels, one Long Barrel (LB) and two
Extended Barrels (EB), each barrel consisting of 64 identical modules;
so the reader is expecting 3x64 = 192 modules;
if the longer LB is indeed done by two identical sub-sections (in case I guess connected at eta = 0), so that the total number of modules is
4x64 = 256, it should be specified for a more clear reading;
- line 54-55: ‘around 200 simultaneous proton-proton collisions will be produced in every bunch crossing’ -> the number of p-p interactions per bunch crossing is fluctuating event by event, so it should be specified if 200 is around the average number or the maximum number of expected p-p interactions per bunch crossing;
Best Regards
Author Response
Dear Reviewer,
many thanks for the careful reading of my article. I have modified the text according to your suggestions.
Best regards
Reviewer 3 Report
Overview: the author presents a clear overview of the ATLAS TileCal upgrade, a critical component of the ATLAS upgrade for the HL-LHC. An overview of the TileCal is given, studies of aging, readout electronics, HV electronics and the results of dedicated test beam experiments. The paper is broad and detailed enough to follow the main elements of the TileCal upgrade program. The language is generally clear.
There are a few sections describing figures that should add some detail, and there are a few cases with the big picture that seems to be missing. I elaborate on these points below. My recommendation is to publish with minor revisions for clarification.
Broad comments:
Structure: it would be helpful if after the introduction some layout of what will come in the text were given.
Overview of trigger system: there seems to be some mixing of the Run 3 and Run 4+ (phase II) trigger systems. In the abstract, the readout rate is stated as 1 MHz, while in L44 it is stated as 100 kHz. The difference seems to be originating from a description of the current system vs. the planned system, but the difference is not transparent to a non-expert reader. I recommend adding a paragraph explaining the evolution of the trigger system, perhaps near the current L44. Another minor point is that the first-level trigger is described simply as L1 uniformly, while some figures mention L0 (see Figure 3/4).
Another issue that is not apparent is the granularity of tile information that will be sent to the L1 trigger system. The current text seems to describe the Run 3 system, while the Run 4 system will have significantly more granularity, which should be stated clearly.
Physics case: While L53ff describes the hardware case (e.g., radiation damage, TDAQ) the reader is left wondering what the physics impact of the upgrade will have. The direct impact of the upgrades on physics may not be particularly transparent. However, at least one minimal suggestion would be to add a few sentences outlining the broad physics goals of the HL-LHC program at ATLAS, such as diHiggs searches. These require higher rate triggers, which is clearly described in the text. With such a small change the link between physics and the hardware upgrade would be more transparent.
Figure 2 / L103ff. There are some obvious questions about this plot that are not explained, such s why there is a discontinuity of data for 250-300 C of charge (top line) and 150-200 C (bottom line). The figure also does not explain what the different data series represents. What are the models used? The text in this section needs to explain the details of how this plot was made as well.
Figure 7 / L 253ff. Similarly to Figure 2, it seems some information was not included in the description. It should be made clear in the caption that the beam is test beam data. Fits of data and simulation are also shown, but the fit function is not described. Finally — regarding the conclusion of the test beam data, is it possible to put the uncertainty on the energy response in perspective, e.g., compared to Run 2? This is not at all a requirement but would be nice to understand if such a comparison can simply be made.
Specific (and very minor) comments
L23: Tile Calorimeter (Tile Cal) should be defined in the text of the paper.
L48: the statement of 4ab-1 is larger than what has typically been stated in the past, which is 3ab-1. This deserves a reference beyond 1705.08830, which shows in figure 1 an integrated luminosity of 3ab-1.
L171: fully finalized ->finalized (unclear what the difference between final and fully final would be)
Author Response
Dear Reviewer,
thank you very much for the careful reading of my article. I have included all of your suggestions into the new version.
The only exception is the wish to put the uncertainty in the energy response in perspective. It is rather difficult to do in a short proceedings, but we will attempt to address this in the next publication of test-beam studies
Best regards
Reviewer 4 Report
The paper describes an important upgrade for the ATLAS detector to its TileCal component. I have a limited number of comments which should be straightforward to address and a few which may be worth considering for future work.
Immediate comments:
1. Line 108. "the study of PMT response" -- is this a dedicated special study or a selected set of in-situ PMTs running in a special condition? Under what conditions was the test performed? If there is not sufficient space here to provide reasonable detail, perhaps a reference could be made to describe the test program more fully.
2. Section 4.1.1 -- is the FENICS electronics COTS or custom? If custom, is there a more-complete design reference?
3. Section 4.1.3 -- are these ADCs custom or COTS? If COTS, this information would be valuable to other researchers. If they are custom, a reference would be appropriate.
4. Line 171 -- English needs work on this line
5. Figure 7. Plots are very difficult to read, particularly because the first and last data points are too close to the Y axes. In addition, every effort should be made to make the text (axis labels, legend) larger and avoid unused whitespace.
Longer-term comment:
Section 7. The analysis of linearity and particle response is valuable. However, the results do not demonstrate very clearly the various key performance aspects of the detector. Indeed, the comparison between pions, kaons, and protons is typically mostly a question of the GEANT4 modelling and physics lists, which are not the topic of the current paper.
Instead, topics of concern would be:
(a) Clear demonstration of linking the linearity of response to the appropriate behavior of the FENIX front-end over both gain channels (e.g. intercalibration)
(b) A study of tails in the data. With continuous readout, it should be possible to accumulate truly samples to demonstrate that there are no anomalous large signals produced in the electronics chain. This could (and perhaps was) be done in radiation tests as well as in testbeam. Such fake signals can be a leading source of trigger issues (e.g. CMS ECAL spikes).
Author Response
Dear Reviewer,
thank you very much for the careful reading of my article. I have included all of your suggestions into the new version.
Here are detailed answers to some of your comments:
1. L108: I have provided references to the existing talks and proceedings. This is an on-going study, so we don't have a solid publication in a peer-review journal yet.
2. and 3. : The TileCal upgrade is based on COTS. That was a main requirement for us. I added some clarifications in the text.
5. I enlarged the size of the plot. Unfortunately, I cannot modify the axis labels, since it would have to go via the approval procedure in our collaboration.
Long-term: I really appreciate your suggestions. We will definitely take them into consideration for our future test-beam campaigns.
Best regards